# Intra-Pulse Modulation Recognition of Radar Signals Based on Efficient Cross-Scale Aware Network

**DOI:** 10.3390/s24165344

**Published:** 2024-08-18

**Authors:** Jingyue Liang, Zhongtao Luo, Renlong Liao

**Affiliations:** 1Hunan Nanoradar Science and Technology Co., Ltd., Changsha 410205, China; jyliang@nanoradar.cn; 2School of Communication and Information Engineering, Chongqing University of Posts and Telecommunications, Chongqing 400065, China; roxlong@foxmail.com

**Keywords:** intra-pulse modulation recognition, convolutional neural network (CNN), time–frequency images (TFIs), cross-scale aware (CSA)

## Abstract

Radar signal intra-pulse modulation recognition can be addressed with convolutional neural networks (CNNs) and time–frequency images (TFIs). However, current CNNs have high computational complexity and do not perform well in low-signal-to-noise ratio (SNR) scenarios. In this paper, we propose a lightweight CNN known as the cross-scale aware network (CSANet) to recognize intra-pulse modulation based on three types of TFIs. The cross-scale aware (CSA) module, designed as a residual and parallel architecture, comprises a depthwise dilated convolution group (DDConv Group), a cross-channel interaction (CCI) mechanism, and spatial information focus (SIF). DDConv Group produces multiple-scale features with a dynamic receptive field, CCI fuses the features and mitigates noise in multiple channels, and SIF is aware of the cross-scale details of TFI structures. Furthermore, we develop a novel time–frequency fusion (TFF) feature based on three types of TFIs by employing image preprocessing techniques, i.e., adaptive binarization, morphological processing, and feature fusion. Experiments demonstrate that CSANet achieves higher accuracy with our TFF compared to other TFIs. Meanwhile, CSANet outperforms cutting-edge networks across twelve radar signal datasets, providing an efficient solution for high-precision recognition in low-SNR scenarios.

## 1. Introduction

Radar signal modulation recognition can be classified into two categories: inter-pulse modulation recognition and intra-pulse modulation recognition [1]. Early studies focused on inter-pulse characteristics for signal recognition [2,3,4,5]. With the advancement of radar technology and the increasing complexity of radar systems, traditional inter-pulse feature analysis becomes insufficient [6]. In recent years, intra-pulse feature analysis has attracted growing attention and has become a valuable tool in the field of radar signal recognition, offering significant advantages in terms of recognition efficiency and accuracy [7].

The traditional way of intra-pulse modulation recognition is based on manual feature design and pattern matching. This approach has some shortcomings, e.g., the complexity of feature extraction, the limitation to single signal recognition, the inadequacy in terms of efficiency, and the poor performance under low-SNR conditions [8]. With the advancement of deep learning (DL), signal recognition based on neural networks has become a promising solution for intra-pulse modulation recognition, as it offers the potential for intelligently recognizing complex, multi-class radar signals [9,10,11,12,13].

Generally, intra-pulse modulation recognition based on DL uses either the signal sequences or the signal time–frequency images (TFIs) as its input. Employing the signal sequence as input means extracting the intra-pulse characteristics using the designed network. For instance, ref. [14] proposes a modified convolutional neural network that uses the signal sequence as input. Ref. [15] designs its algorithm by combining a convolutional neural network (CNN) with a long short-term memory (LSTM) network, which also adopts signal sequences as input. In [16], an omni-dimensional dynamic-convolution-layer-based network (OD-CNN) with a focal loss function is designed and applied to classify radar intra-pulse modulations based on signal sequences.

In contrast, using the TFIs as input implies the signal is preprocessed before the network. In [17], the Choi–Williams distribution (CWD) is employed as an input to an improved deep residual network (ResNet). Ref. [18] transforms the radar signal’s bicubic interpolation Wigner–Ville distribution (WVD) matrix into a square matrix. This matrix is then utilized to train a CNN for signal recognition.

However, there are two primary challenges to improving the performance of intra-pulse modulation recognition using TFIs and CNNs. The first challenge is the severe contamination of TFIs at low SNRs. Reference [19] employs an improved convolutional denoising autoencoder (CDAE) to de-noise TFIs and then utilizes a CNN to identify 10 radar signals at a −6 dB SNR, achieving a recognition rate exceeding 88%. However, this approach requires an additional noise estimation network and does not extend its application to lower-SNR scenarios.

The second challenge lies in the limitations of CNNs to effectively extract features [20], particularly due to the local characteristics of convolutional layers, which makes it difficult for them to capture global information [21]. To overcome this, some researchers have proposed increasing the number of layers within CNNs, allowing for the extraction of more complex features [22]. However, adding more layers introduces redundancy into the network and, consequently, increases computational time and complexity.

This paper proposes to design a lightweight CNN that makes use of multiple TFIs jointly. The key challenge is how to extract signal features comprehensively under low SNRs, since the contextual information carried by the TFIs differs in amount, range, etc., for various modulation types of radar signals. A transformer is able to address contextual information, but it generally requires a large number of network parameters [23]. In this paper, to solve this problem, we use a combination of three different types of TFIs. They have more nonchiasmatic information, which complements the defects between them. A series of meticulous preprocessing techniques are used for noise reduction and image sizing. Then, channel fusion technology is utilized to fuse TFIs to form a time–frequency fusion feature (TFF) as the object of deep neural network learning.

In this paper, we propose a lightweight cross-scale aware network (CSANet) to recognize the modes of intra-pulse modulation based on TFIs; it consists of cross-scale aware (CSA) modules, convolution layers, and fully connected layers. The CSA module employs spatial and channel attention mechanisms on feature blocks across different scales and has a residual structure to avoid the problems of exploding and vanishing gradients. Furthermore, to ensure the network remains lightweight and computationally efficient, we consider multiple aspects to design the CSA module, including depthwise convolution, a parallel branch architecture, and channel size adjustment. Experiments demonstrate that the CSA module can direct the attention of the CNN towards global features while also recognizing the time–frequency structure of radar signals.

## 2. Signal Model and System Overview

### 2.1. Signal Model

Intra-pulse modulation modes of radar signals mainly include frequency modulation, phase modulation, combined modulation, etc. [24]. Table 1 shows the 12 typical modulation modes of radar signals that are used in this paper, where *A*, *T*, and $f_{c}$ denote the amplitude, pulse width, and carrier frequency, respectively, *B* is the bandwidth, *k* is the slope of the frequency modulation, φ is the primary phase, Δf denotes the frequency interval, *c* is a random code that controls the frequency modulation, *N* is the number of codes, $T_{s}$ is the width of a code, and *M* is the count of sub-pulses within one group. Further, $g_{T}\left(t\right)=1/\sqrt{T}\mathrm{rec}\left(t/T\right)$, where the rectangular function $\mathrm{rec}(t^{\prime })=1$, and $t^{\prime }\in \left[0,1\right]$.

Considering a noisy environment, we model the received signal as
(1)y(t)=x(t)+n(t)
where y(t) is the received signal, x(t) is the radar signal, and n(t) is the additive noise, which is usually considered as white Gaussian noise.

### 2.2. System Overview

In this paper, we design an intra-pulse modulation recognition system for radar signals that consists of feature extraction and a CSANet classifier, as illustrated in Figure 1. Our system contains three primary steps:

(1) Time–frequency analysis: Initially, we apply time–frequency analysis techniques to the radar signals to obtain the TFIs. Specifically, this paper utilizes three distinct types of time–frequency features: FSST (Fourier synchrosqueezed transform) [25], *SPWVD* (smoothed pseudo Wigner–Ville distribution) [26], and HHT (Hilbert–Huang transform) [27].

(2) Image preprocessing: Subsequently, image preprocessing approaches, including binarization and cubic interpolation clipping, are conducted on the TFIs. Then, the obtained time–frequency features are fused into the TFF feature.

(3) Feature fusion and model training: Finally, we construct TFF feature datasets from various signals and scenarios, which are divided into training and test sets for the CSANet. The CSANet is applied to recognize the 12 types of radar signals.

In the following, Section 3 presents time–frequency analysis and the feature extraction process, and Section 4 details the architecture of our CSANet.

## 3. Time–Frequency Analysis and Feature Extraction

This section discusses time–frequency analysis techniques and the feature extraction process. Section 3.1, Section 3.2 and Section 3.3 introduce three time–frequency analysis techniques, respectively. Section 3.4 presents the methods of TFI preprocessing and TFI fusion.

### 3.1. Cohen Class Time–Frequency Distribution

As a typical case of a quadratic time–frequency distribution, a Cohen class time–frequency distribution typically employs a kernel function to smooth the quadratic function of signals [28]. This process requires a balance between time–frequency resolution and a cross-term. The Cohen class time–frequency distribution can be formulated as
(2)$$\begin{array}{cc} C(t,f) & =\frac{1}{4\pi^{2}}\int_{-\infty }^{+\infty }\int_{-\infty }^{+\infty }\int_{-\infty }^{+\infty }x\left(u+\frac{\tau }{2}\right)x^{*}\left(u-\frac{\tau }{2}\right)\phi \left(\tau ,v\right)e^{-j2\pi (tv-uv+f\tau )}\;dud\tau dv \end{array}$$
where *t*, *f*, τ, *v*, and *u* denote the time, frequency, time delay, frequency shift, and center of the correlation function, respectively, and ϕ(τ,v) represents the kernel function. Different kernel functions lead to different kinds of Cohen class time–frequency distributions.

The Cohen class time–frequency distribution is equivalent to the Wigner–Ville distribution (WVD) when ϕ(τ,v)=1. The WVD is characterized by its high time–frequency resolution. However, when the signal comprises multiple components, the WVD can produce cross-terms. Interference among the signal’s components can result in the mixing of their characteristics, potentially obscuring the distinct features of the individual components [29].

A smoothed pseudo-Wigner–Ville distribution (*SPWVD*) suppresses the cross-term interference by smoothing the WVD with two window functions. One window function operates in the time domain, while the other is applied in the frequency domain. A *SPWVD* can be formulated as
(3)$$\begin{array}{cc} SPWVD(t,f) & =\int_{-\infty }^{+\infty }h(\tau )\int_{-\infty }^{+\infty }g(u-\tau )x\left(u+\frac{\tau }{2}\right)x^{*}\left(u-\frac{\tau }{2}\right)e^{-j2\pi t}dud\tau \end{array}$$
where h(τ) and g(u) are the smoothing window functions.

The *SPWVD* effectively seeks a balance between the high time–frequency resolution and the suppression of cross-term interference. Figure 2 shows the *SPWVD* of radar signals with various modulation modes when SNR = 10 dB. As can be seen, each modulation mode exhibits distinctive behavior.

### 3.2. Fourier Synchrosqueezed Transform (FSST)

The short-time Fourier transform (*STFT*) is formulated as
(4)$$STFT\left(t,f\right)=\int_{-\infty }^{+\infty }x(\tau )\eta^{*}\left(\tau -t\right)e^{-j2\pi f\tau }d\tau$$
where ${\left(\cdot \right)}^{*}$ denotes the conjugate of a complex number, and η denotes the window function. Based on the *STFT*, the FSST is formed by a synchronous compression transform, defined as
(5)$$T_{f}\left(t,\omega \right)=\frac{1}{\eta^{*}\left(0\right)}\int_{-\infty }^{+\infty }STFT\left(t,f\right)\delta \left[\omega -{\hat{\omega }}_{f}\left(t,f\right)\right]df$$
where ω represents the frequency of the correction function, and δ is the impulse response. The term ${\hat{\omega }}_{f}\left(t,f\right)$ is the local instantaneous frequency, given by
(6)ω^f(t,f)=Re1j2π∂tSTFT(t,f)STFT(t,f)
where $\partial_{t}$ denotes the partial derivative, and Re(·) denotes the real part. FSST compresses the time–frequency curve along the frequency dimension, thereby concentrating the signal energy in the time–frequency domain. This concentration effectively minimizes the noise’s impact. One example is presented in Figure 3.

### 3.3. Hilbert–Huang Transform (HHT)

HHT combines the Hilbert transform and adaptive signal decomposition to form a time–frequency feature. For instance, the empirical mode decomposition (EMD) or the variational mode decomposition (VMD) is employed to decompose radar signals into a collection of sub-signals, i.e., intrinsic mode functions (IMFs) [30]. Subsequently, the Hilbert method is utilized to derive time–frequency characteristics. Unlike EMD, VMD is a non-iterative signal processing technique. By iteratively searching for the optimal solution of the variational modes, VMD refines the optimal central frequencies and bandwidths of the IMFs adaptively. Therefore, VMD is much more robust to sampling and noise than EMD [31].

VMD processing includes two steps. Firstly, based on the input signal x(t), the set of IMFs $u_{k}$ is calculated by the decomposition algorithms [32]. Then, for each IMF, its Hilbert transform is calculated as
(7)$$d_{i}\left(t\right)=\frac{1}{\pi }\int_{-\infty }^{+\infty }\frac{u_{i}\left(\tau \right)}{t-\tau }d\tau .$$

Hence, the instantaneous frequency is
(8)$$\omega_{i}\left(t\right)=d\left(\frac{d_{i}\left(t\right)}{u_{i}\left(t\right)}\right)/dt.$$

Figure 4 shows one example of an HHT.

### 3.4. Time–Frequency Feature Preprocessing and Fusion

Using the FSST, *SPWVD*, and HHT methods, we obtain three types of time–frequency images. It is necessary to preprocess these images for more accurate identification. At present, there are two types of methods: image reconstruction based on neural networks and denoising based on traditional signal processing technology. However, some methods may not perform well under low-SNR conditions. For example, we employ CDAE to process *SPWVD*, and we use different noise variances to train the CDAE to reconstruct images [19]. The results are shown in Figure 5.

We introduce an image preprocessing step designed to reduce the impact of noise and the computational complexity for deep neural networks. As depicted in Figure 6, the preprocessing process encompasses the following steps: (a) converting the original TFIs to grayscale, (b) applying adaptive threshold binarization [33], (c) employing morphological operations to fill in missing points and remove noise-induced outliers, and (d) employing bicubic interpolation to resize the images to 256×256 pixels to make them suitable for input into the CSANet mode. This series of preprocessing filters out a lot of noise and mainly preserves the outline of the time–frequency ridge, which is the key to radar signal recognition [34].

Given that *SPWVD*, FSST, and HHT are based on different principles and show different time–frequency distributions, the use of feature fusion to enhance feature extraction performance, especially at low SNRs, is a promising approach. *SPWVD* belongs to the Cohen class of time–frequency distributions and is a quadratic, nonlinear time–frequency distribution with high time–frequency resolution, but it is susceptible to the influence of cross-terms. FSST is a linear time–frequency distribution based on the time window, which enhances the time–frequency concentration of *STFT* through the synchronous compression operator. However, due to the fixed window and basis function in the analysis, it performs poorly in matching multi-component and time-varying signals. HHT consists of two steps: variational mode decomposition (VMD) and Hilbert amplitude spectrum analysis (HAS). It is fully adaptive and is capable of processing nonlinear and non-stationary data, but it suffers from the issue of mode aliasing. These three types of time–frequency distributions each have their advantages and belong to different categories, possessing non-cross information. Therefore, combining these three types of time–frequency analyses can enhance the robustness of the integrated features.

After the preprocessing of three classes of time–frequency images, we construct the TFF as a multi-channel two-dimensional image by concatenating the images along the channel dimension. Although image fusion increases the computational load for both the time–frequency analysis and the initial layer of the neural network, the TFF has been demonstrated to effectively enhance recognition performance.

## 4. Cross-Scale Aware Network (CSANet)

After feature extraction, we need to design a network to recognize the type of radar signal modulation. CNN, as a classical type of neural network proposed by Yann LeCun [35], has been widely used for radar signal modulation recognition. To improve the accuracy of signal modulation recognition under low SNRs, complex deep CNNs have been employed in recent years. For instance, ResNet, which uses the residual structure and easily constructs networks with dozens or even hundreds of layers [36], performs well in recognizing radar signals. However, there is a need for further improvement in recognition accuracy. Additionally, complex networks often face difficulties when applied to lightweight platforms due to their computational demands.

In this paper, we design CSANet, which offers high recognition accuracy and low computational complexity. The CSANet architecture is presented in Figure 7a. CSANet extracts the TFF image features using four convolution (Conv) layers, two maximum pooling (MP) layers, and three CSA modules. Then, the extracted features are flattened and connected to a linear layer, and classification results are obtained. The CSA module is an essential component of the proposed CSANet. The following details the operation process of the CSA module and its constituent components.

### 4.1. CSA Module

Figure 7b depicts the operation process of the CSA module. As can be seen, the CSA module employs residual connections and conducts multi-branch feature extraction using the DDConv Group with a parallel structure. It then fuses multiple time–frequency distributions through a cross-channel interaction (CCI) and recognizes the time–frequency structural characteristics of radar signals by spatial information focus (SIF). Finally, it integrates the information through a nonlinear gated fusion unit (GFU).

### 4.2. Depthwise Dilated Convolution Group (DDConv Group)

Figure 7c shows the structure of the DDConv Group, where DDC represents the depthwise dilated convolution. The expansion rates of the four branches are 1, 3, 5 and 8, respectively. The DDConv Group is used to extract multi-scale features. Instead of the commonly used depthwise separable convolution, we utilize a 3 × 3 depthwise convolution; each channel operates with an independent convolution kernel, which significantly reduces the computational requirements. The 1 × 1 convolution in the depthwise separable convolution can reduce the dimensions and carryout channel flow, but dimensionality reduction is not conducive to feature retention. Therefore, we complete the work of channel flow by our designed CCI. Meanwhile, through different receptive fields, SIF can capture the global dependence in the feature information and has advantages over modulated signals with complex time–frequency energy distributions, e.g., polyphase coding and multi-frequency coding signals.

### 4.3. Cross-Channel Interaction (CCI)

Channel attention is one of the attention mechanism types. An example of channel attention is the squeeze-and-excitation network (SENet) [37]. Generally, channel attention compresses the input feature map from the channel direction, generating weights for each channel to represent the importance of the current channel. In this way, the model can focus on the more important channels, thereby improving performance.

Our designed CCI module aims to strength the integration of TFF characteristics, extract non-overlapping information from three time–frequency distributions, and suppress the noisy channels. CCI applies the channel attention mechanism to each scale of the feature branch. For the DDConv Group, it provides the multi-scale feature $X_{i}\in R^{C\times H\times W}$ to the CCI, where i=[0,1,2,3] denote different scale feature blocks.

Figure 7d depicts the operation process of the CCI. First, a global feature representation is obtained through a spatial global average pooling (SGAP), which works by averaging the two-dimensional feature map of each channel. Then, 1 × 1 convolution (Conv) is used to model the inter-channel relationships. The sigmoid activation function is employed to generate the channel descriptor, and the softmax function is utilized to obtain the representation weight, which is given by
(9)$$\begin{array}{c} W_{i,CH}=\mathrm{softmax}\left\{\mathrm{sigmoid}\left[\mathrm{Conv}(\mathrm{SGAP}\left(X_{i}\right))\right]\right\}\in R^{C\times 1\times 1} \end{array}$$
where $\mathrm{softmax}\left\{\mathrm{sigmoid}\left[\mathrm{Conv}(\mathrm{SGAP}(\cdot ))\right]\right\}$ is defined as $F_{1}\left(\cdot \right)$. By performing element-wise multiplication of the weights and descriptors, we have the multi-scale fusion feature $Y_{CH}\in R^{C\times H\times W}$ as
(10)$$Y_{CH}=\sum_{i=0}^{3}W_{i,CH}⊙X_{i}$$
where ⊙ denotes the Hadamard product.

### 4.4. Spatial Information Focus (SIF)

Channel attention focuses on the differences in features in different channels, while spatial attention emphasizes the information in different locations of the image [38]. Basically, spatial attention learns a spatial transformation matrix that is used to transform the input feature map into a new feature map wherein key information is highlighted and irrelevant information is suppressed. This mechanism helps the model to focus more on important spatial locations within the image, thereby enhancing the performance of the network model.

In this paper, SIF is designed as a parallel branch to CCI. While CCI pools spatial information at different scales to compute the channel attention descriptor, SIF requires pooling in the channel dimension. Therefore, to avoid losing information, SIF fuses the multi-scale features $X_{i}\in R^{C\times H\times W}$ in the channel dimension and then outputs $\left[X_{F}\in R^{4C\times H\times W}\right]$. Figure 7e shows the operation process of SIF.

Considering that the channel dimension is a one-dimensional feature and has fewer parameters during global pooling, we design SIF with CGAP (channel global average pooling) and CGMP (channel global maximum pooling) to obtain global features and then fuse them in the channel dimension, given by
(11)$$\begin{array}{c} X_{A}=\mathrm{CGAP}\left(X_{F}\right)\in R^{1\times H\times W},\\ X_{M}=\mathrm{CGMP}\left(X_{F}\right)\in R^{1\times H\times W},\\ X_{AM}=\mathrm{Cat}\left(X_{A},X_{M}\right)\in R^{2\times H\times W} \end{array}$$
where Cat(·) denotes the feature map fusion in the channel dimension. A 7 × 7 convolution is used to map the dual-channel $X_{AM}$ into four channels, corresponding to the four scales of the input. The sigmoid activation function is employed to generate the spatial weight representation:(12)$$W_{SP}=\mathrm{sigmoid}\left(\mathrm{Conv}\left(X_{AM}\right)\right)\in R^{4\times H\times W}$$
where sigmoid(Conv(·)) is defined as $F_{2}\left(\cdot \right)$. Then, the weights $W_{i,SP}\in R^{1\times H\times W}$ are from $W_{SP}\in R^{4\times H\times W}$ and are used to compute the multi-scale spatially fused feature $Y_{SP}\in R^{C\times H\times W}$, given by
(13)$$Y_{SP}=\sum_{i=0}^{3}W_{i,SP}⊙X_{i}.$$

### 4.5. Gated Fusion Unit (GFU)

The gated fusion unit (GFU), as depicted in Figure 7f, generates adaptive weights to fuse the outputs of the CCI and SIF branches by the Sigmoid activation function in order to restore the original scale size and improve the feature representation. Given $Y_{CH}\in R^{C\times H\times W}$ and $Y_{SP}\in R^{C\times H\times W}$, the representative weights $Z\in R^{C\times H\times W}$ are calculated as
(14)$$Z=\mathrm{sigmoid}(Y_{CH}W_{1}+Y_{\mathrm{SP}}W_{2})$$
where $W_{1},W_{2}\in R^{C\times C}$ are the learnable parameters during CSANet training. Then, the cross-scale aware feature is achieved by
(15)$$Y_{CSA}=Z⊙Y_{CH}+\left(1-Z\right)⊙Y_{SP}.$$

## 5. Experimental Results

The dataset simulates 12 modulation types of radar signals, as shown in Table 1. The parameters of the signals are set as in Table 2. The sampling frequency is 200 MHz, and the signal length is 10 μs. The SNR is set as [−14,−12,⋯,8] dB. Therefore, for every type of modulation at each SNR point, we construct 350 training samples, 150 validation samples, and 150 test samples. Moreover, our dataset contains 54,600, 23,400, and 23,400 samples for training, validating, and testing, respectively. The experiments are performed using the PyTorch 2.2.1 framework and an NVIDIA GeForce RTX 4060 laptop GPU.

In addition, in order to ensure the comparability and statistical significance of the experimental results, the experimental parameters are uniformly set as follows: the initial learning rate is 0.01, the batch size is 50, the optimization algorithm is stochastic gradient descent, the number of epochs is 50, and the loss function is cross-entropy loss. Through careful adjustment of datasets and parameters, the loss value of each network can reach the convergence state after 50 rounds of training so as to ensure the correct evaluation of the performance of the proposed algorithm under different SNR and parameter conditions.

### 5.1. Accuracy Analysis of CSANet and Other Networks

First, we evaluate the recognition performance of CSANet. For comparison, we simulate five algorithms that are based on the TFIs-CNN methodology, including CNNQu [39], CNNHuang [32], ResNet50 [36], MobileNetV3 [40], and ShuffleNetV2 [41]. To demonstrate the effectiveness of the TFF, we also generate TFIs, i.e., *SPWVD*, FSST, or HHT, as the input of networks. The experimental results are shown in Figure 8.

As can be seen, the accuracy of ResNet50, MobileNetV3, ShuffleNetV2, and CSANet is higher when TFF is used as the learning object, while for CNNQu and CNNHuang, the accuracy is higher by inputting FSST. This demonstrates that TFF is superior to the time–frequency enhancement methods [25,26,27].

Overall, CSANet-TFF consistently achieves the highest accuracy across all SNR levels. Notably, at a low SNR of −14 dB, CSANet-TFF attains the highest accuracy of 83.62%, which exceeds the second-highest accuracy of ResNet50-TFF by 8.74%. At SNR = −12 dB, the accuracy of CSANet-TFF is 93.27%, outperforming other networks. Generally, CSANet-TFF excels in low-SNR scenarios. Moreover, CSANet consistently achieves the highest accuracy with *SPWVD*, FSST, HHT, or TFF as inputs, demonstrating that it outperforms existing advanced methods [32,36,39,40,41]. This superiority is attributed to CSANet’s ability to perceive the time–frequency structures of radar signals through the CSA module. The DDConv Group and SIF mechanisms within the module are specifically designed to identify time–frequency ridge features, while CCI is designed to suppress redundant channels caused by noise, particularly in low-SNR conditions.

Furthermore, we compare the network across several metrics, including spatial complexity, computational complexity, and actual running time. These comparisons are based on parameters (Params), floating-point operations (FLOPs), and network inference time (Runtime), with the results presented in Table 3. As can be seen, CNNQu and CNNHuang exhibit lower FLOPs but higher Params compared to CSANet. However, as illustrated in Figure 8, their accuracy is significantly lower than that of CSANet, particularly at low SNRs. CSANet has a more lightweight architecture, with reduced FLOPs and Params compared to ResNet50. Moreover, CSANet’s Params are approximately 1/10 of those of MobileNetV3 and ShuffleNetV2. Therefore, our proposed CSANet is proven to be a lightweight network with high accuracy. The inference time of the network is often affected by the hardware resources. Here, we calculate the time for different networks to classify a single sample, which provides a reference for the actual deployment of the network. Experimental results show that CSANet can achieve a shorter time than [36,40,41], which has practical application significance.

### 5.2. Ablation Study

This paper designs the CSA module that enables the CSANet to recognize intra-pulse modulation of radar signals. Here, we conduct an ablation study by adding the CSA modules one by one into the CSANet. When one CSA module is added, we employ 3×3 convolutions with a stride of two for further feature mapping and downsampling so as to minimize the data redundancy and forge an efficient architecture for CSANet. When the number of CSA modules grows, CSANet keeps its overall structure and adjusts the parameter number of the “Linear” layer accordingly. Furthermore, ablation experiments are carried out on the internal modules. SIF and GFU are removed, and the remaining modules are named D-CCI. CCI and GFU are removed, and the remaining modules are named D-SIF. We use D-CCI and D-SIF to replace three CSA modules in CSANet. In addition, we contrast CBAM [42], which has a dual-channel and spatial attention mechanism, with some similarities to CSA. In the experiment, we use a CBAM module to replace three CSA modules in CSANet. Using the TFF features as learning objects, the results of the ablation experiments are illustrated in Table 4.

CBAM is less effective although it exerts both spatial and channel attention, reflecting the importance of multi-scale features. At the same time, D-SIF and D-CCI also have lower recognition accuracies than the CSA = 3 architectures, but they require slightly less computational effort. D-CCI demonstrates superior performance due to the utilization of integrated features, which encompass comprehensive and even redundant information. Therefore, the suppression of channel redundancy is critical. CSANet’s accuracy increases when the CSA number grows from one to three, but it declines when the CSA number reaches four. Due to the deployment of convolutional layers for downsampling following each CSA to enhance network efficiency, an increased number of CSA modules may lead to greater information loss. When the number of CSA modules is less than three, the information loss predominantly consists of redundant data caused by noise. Conversely, when the number of CSA modules exceeds four, there is a consequential loss of useful information.

### 5.3. Signal Confusion Analysis

This section analyzes the confusion matrix of the CSANet results in order to provide insights into the model’s performance on different modulation types. Figure 9 depicts the confusion matrices of CSANet based on three methods: *SPWVD*, FSST, and TFF at SNR = −12 dB. HHT is ignored since its performance is the worst.

In Figure 9, the vertical axis lists the true labels, while the horizontal axis lists the predicted labels. The diagonal elements represent the number of samples correctly predicted, the other elements represent the number of samples incorrectly classified, and the darker the color, the more samples. In Figure 9, three time–frequency features and twelve modulation types show excellent discrimination. It is relatively easy to confuse BPSK versus CW and FRANK versus P3. In addition, at very low SNRs, most of the signals are misinterpreted as 4FSK signals because their time–frequency graphs are irregular and scattered.

Figure 10 shows the accuracy of CSANet for the recognition of twelve modulations based on *SPWVD*, FSST, and TFF. As can be seen, the FSST’s accuracy varies significantly across different signal types. The FSST performs well in recognizing LFM, NLFM, FSK, and P4, but it performs poorly in recognizing FRANK, P3, and FSK4, even for high SNRs. Hence, the FSST’s performance is sensitive to the modulation type. *SPWVD* and TFF are robust in recognition of various signals. The recognition accuracies of NLFM, LFM, P4, and P2 are generally higher, and the time–frequency ridges of NLFM and LFM are simpler. P2 and P4 have two ridges at the edges of their time–frequency maps, as shown in Figure 2. Moreover, P2 has phase mutations and P4 does not, and they are also distinguished from each other. Specifically, CSANet-TFF achieves over 94.73% accuracy at SNR = −10 dB for every signal.

### 5.4. Class Activation Mapping (CAM) Analysis

CAM is widely used for explaining the predictions of DL models [43]. CAM helps researchers understand how a DL model can choose the predicted class by mapping the class activation back to the significant region of the image. Figure 11 shows the CAM analysis results of CSANet and ResNet50 with TFF features, where the brighter regions are more important. The CAM analysis aids in understanding the adaptive receptive field of CSANet. We extract the feature maps before the Linear layer for visualization analysis.

As can be seen from Figure 11, the receptive field of CSANet shows a higher degree of concentration than that of ResNet50. For CSANet, the focus area adaptively adjusts in size, shape, and position, corresponding to the characteristics of varying signals, which is due to the DDConv Group and the SIF modules. Combined with the DDConv Group, which employs convolution kernels of varying dilation ratios, SIF can extract features with different ranges and precision.

## 6. Conclusions

In this paper, we propose CSANet, a lightweight and accurate model for recognizing intra-pulse modulation in radar signals. We design TFF using three types of TFIs, i.e., *SPWVD*, FSST, and HHT. In our experiments with 12 radar signal types, CSANet using TFF achieves accuracies of 83.62%, 93.99%, and 98.23% at SNR levels of −14, −12, and −10 dB, respectively.

CSANet’s high precision is mainly attributed to the CSA module, which is specifically designed to effectively address the characteristics of time–frequency ridges, including large spans, narrow curves, and sharp changes. Our solution is to develop a cross-scale strategy that correlates information across different scales and benefits the identification of key features. In the CSA module, the DDConv Group employs multiple dilated convolutions to extract multi-scale feature blocks. Two parallel branches are developed by jointly employing the channel and spatial attention mechanisms to highlight discriminative features and mitigate channel redundancies across various scales.

In terms of network complexity, we employ depthwise dilated convolutions to make the CSA lightweight. Compared to [32] with 21.01 M Params and [39] with 2.55 M Params, CSANet has only 0.22 M Params. Therefore, CSANet is a promising tool for accurate recognition of radar signals, especially in low-SNR conditions.

## Figures and Tables

**Figure 1 sensors-24-05344-f001:**
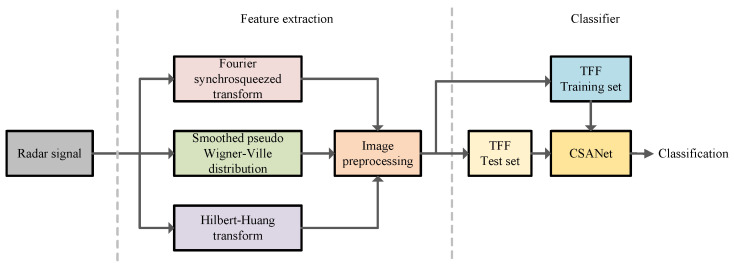
Structure diagram of the recognition system.

**Figure 2 sensors-24-05344-f002:**
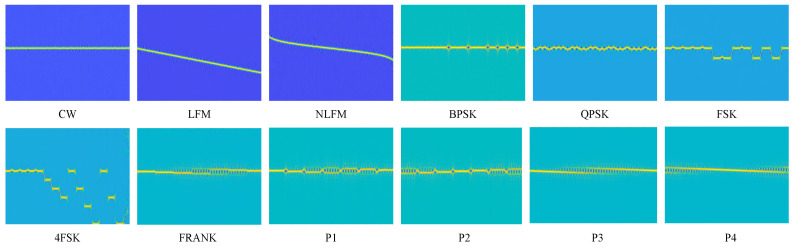
*SPWVD* of various radar signals for SNR = 10 dB.

**Figure 3 sensors-24-05344-f003:**
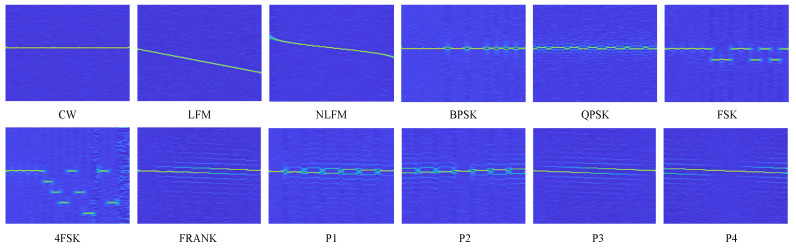
FSSTs of various radar signals for SNR = 10 dB.

**Figure 4 sensors-24-05344-f004:**
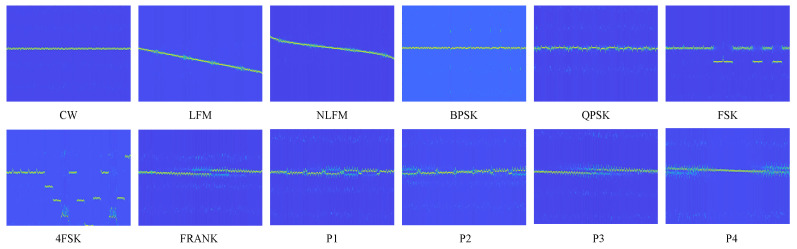
HHTs of various radar signals for SNR = 10 dB.

**Figure 5 sensors-24-05344-f005:**
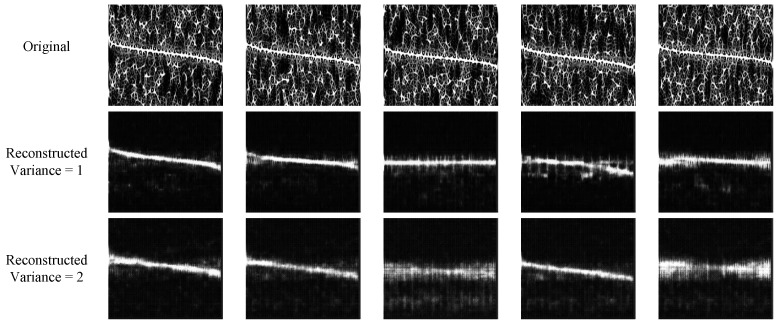
*SPWVD* of NLFM signal reconstructed based on CDAE; SNR = −8 dB.

**Figure 6 sensors-24-05344-f006:**

*SPWVD* image preprocessing of the NLFM signal for SNR = −8 dB.

**Figure 7 sensors-24-05344-f007:**
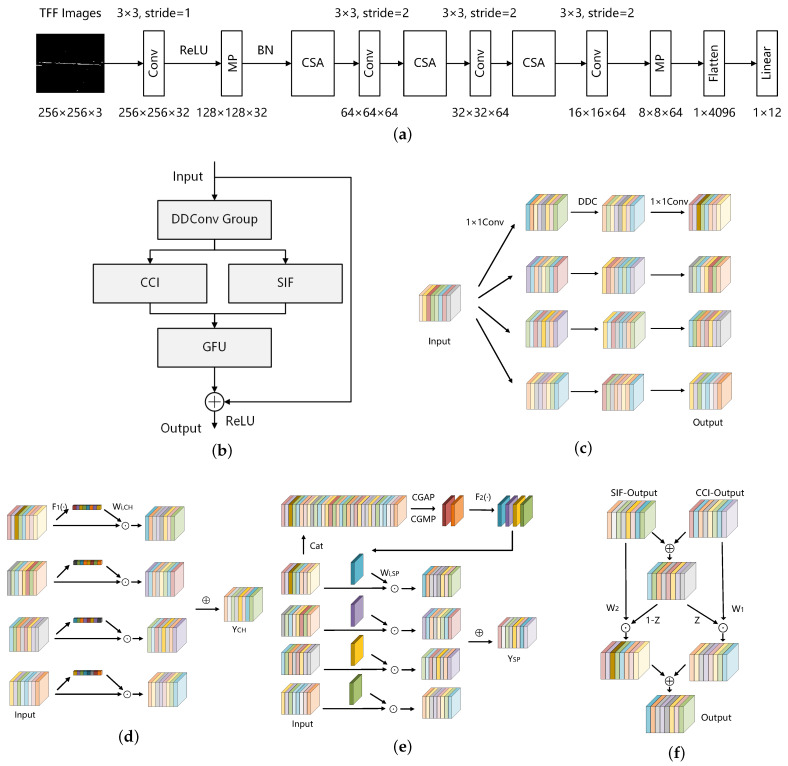
The overall architecture of CSANet: (**a**) CSANet. (**b**) CSA modules. (**c**) DDConv Group module. (**d**) CCI module. (**e**) SIF module. (**f**) GFU module. Multi-scale features are extracted from CSA by DDConv Group, then channel attention is applied by CCI and spatial attention is applied by SIF, and finally, fusion is performed by GFU.

**Figure 8 sensors-24-05344-f008:**
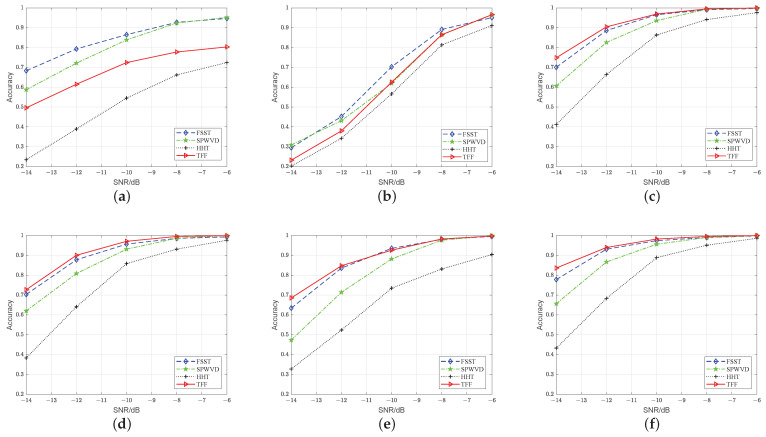
Accuracy evaluation: (**a**) CNNQu, (**b**) CNNHuang, (**c**) ResNet50, (**d**) MobileNetV3, (**e**) ShuffleNetV2 (**b**), and (**f**) CSANet.

**Figure 9 sensors-24-05344-f009:**
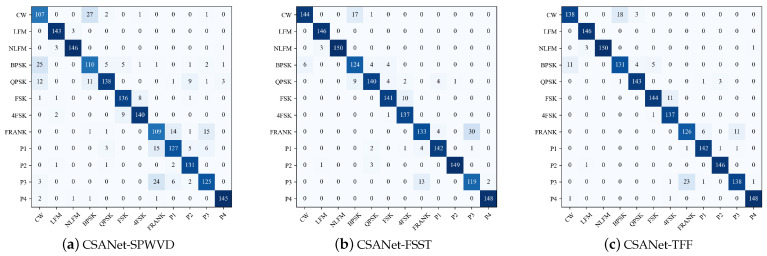
CSANet confusion matrix for SNR = −12 dB: (**a**) *SPWVD*. (**b**) FSST. (**c**) TFF.

**Figure 10 sensors-24-05344-f010:**
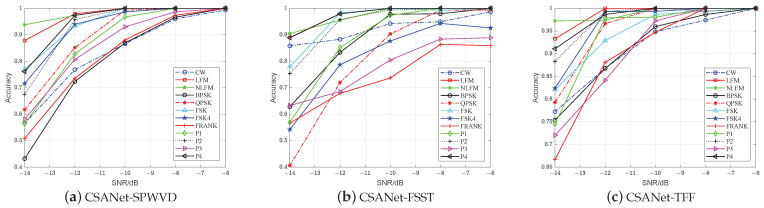
Accuracy of each signal: (**a**) *SPWVD*. (**b**) FSST. (**c**) TFF. The figure illustrates the accuracy of CSANet in identifying signal types using three different sets of time–frequency features to compare the classification effect.

**Figure 11 sensors-24-05344-f011:**
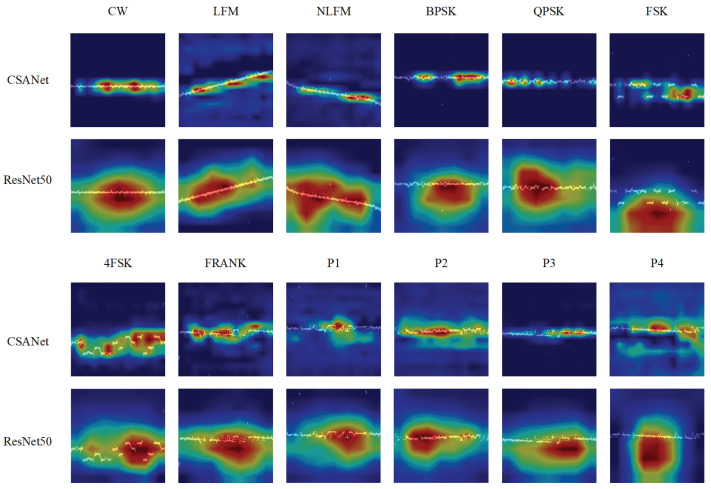
CAMs of CSANet and ResNet50 (SNR = −6 dB).

**Table 1 sensors-24-05344-t001:** The formulas of typical radar signals.

Modulation	Formula
CW (Continuous Wave)	$x\left(t\right)=A\mathrm{rec}\left(t/T\right)e^{j2\pi f_{c}t}$
LFM (Linear Frequency Modulation)	$x\left(t\right)=A\mathrm{rec}\left(t/T\right)e^{j[2\pi \left(f_{c}t+\pi kt^{2}+\varphi \right)]}$
NLFM (Nonlinear Frequency Modulation)	$x\left(t\right)=A\mathrm{rec}\left(t/T\right)e^{j2\pi \int_{0^{-}}^{t}T^{-1}\left(f\right)dt}$
	$T\left(f\right)=T\int_{-\infty }^{f}W\left(u\right)du/\int_{-B/2}^{B/2}W(v)dv,$
	$W\left(f\right)=0.63+0.46\cos\left(2\pi f/B\right)$, $f\in [-\frac{B}{2},\frac{B}{2}]$
BPSK (Binary Phase Shift Keying)	$x\left(t\right)=A\sum_{i=1}^{N}e^{j(2\pi f_{c}t+\varphi )}g_{T_{s}}\left(t-iT_{s}\right)$
	φ=0,π
QPSK (Quadrature Phase Shift Keying)	$x\left(t\right)=A\sum_{i=1}^{N}e^{j(2\pi f_{c}t+\varphi )}g_{T_{s}}\left(t-iT_{s}\right)$
	$\varphi =0,\frac{1\pi }{2},\pi ,\frac{3\pi }{2}$
FSK (Frequency Shift Keying)	$x\left(t\right)=A\sum_{i=1}^{N}e^{j2\pi \left(f_{c}+c\Delta f\right)t}g_{T_{s}}\left(t-iT_{s}\right)$
	c=0,1
4FSK (Four-Frequency Shift Keying)	$x\left(t\right)=A\sum_{i=1}^{N}e^{j2\pi \left(f_{c}+c\Delta f\right)t}g_{T_{s}}\left(t-iT_{s}\right)$
	c=0,1,2,3
FRANK	$x\left(t\right)=A\sum_{i=1}^{N}e^{j(2\pi f_{c}t+\varphi_{i,j})}g_{T_{s}}\left(t-iT_{s}\right)$
	$\varphi_{i,j}=\frac{2\pi }{M}\left(i-1\right)\left(j-1\right),i,j=1,2,\cdots ,M$
P1	$x\left(t\right)=A\sum_{i=1}^{N}e^{j(2\pi f_{c}t+\varphi_{i,j})}g_{T_{s}}\left(t-iT_{s}\right)$
	$\varphi_{i,j}=\frac{-\pi }{M}\left[M-\left(2i-1\right)\right]\left[\left(j-1\right)M+\left(j-1\right)\right]$
	i,j=1,2,⋯,M
P2	$x\left(t\right)=A\sum_{i=1}^{N}e^{j(2\pi f_{c}t+\varphi_{i,j})}g_{T_{s}}\left(t-iT_{s}\right)$
	$\varphi_{i,j}=\frac{\pi }{2M}\left[M+1-2i\right)]\left[M+1-2j\right)]$
	i,j=1,2,⋯,M
P3	$x\left(t\right)=A\sum_{i=1}^{N}e^{j(2\pi f_{c}t+\varphi_{i})}g_{T_{s}}\left(t-iT_{s}\right)$
	$\varphi_{i}=\frac{\pi }{M}{\left(i-1\right)}^{2},i=1,2,\cdots ,M$
P4	$x\left(t\right)=A\sum_{i=1}^{N}e^{j(2\pi f_{c}t+\varphi_{i})}g_{T_{s}}\left(t-iT_{s}\right)$
	$\varphi_{i}=\frac{\pi }{M}{\left(i-1\right)}^{2}-\pi \left(i-1\right),i=1,2,\cdots ,M$

**Table 2 sensors-24-05344-t002:** Simulation parameters for radar signals in Table 1.

Modulation	Parameters and Their Ranges
CW	$f_{c}\in \left[45,55\right]$ MHz
LFM	$f_{c}\in \left[45,55\right]$ MHz, B∈[15,25] MHz
NLFM	$f_{c}\in \left[45,55\right]$ MHz, B∈[15,25] MHz
BPSK	$f_{c}\in \left[45,55\right]$ MHz, N=13
QPSK	$f_{c}\in \left[45,55\right]$ MHz, N=28
FSK	$f_{c}\in \left[45,55\right]$ MHz, Δf∈[10,20], N=13
4FSK	$f_{c}\in \left[45,55\right]$ MHz, Δf∈[5,15], N=16
FRANK	$f_{c}\in \left[45,55\right]$ MHz, N=50, M=7
P1	$f_{c}\in \left[45,55\right]$ MHz, N=50, M=7
P2	$f_{c}\in \left[45,55\right]$ MHz, N=50, M=7
P3	$f_{c}\in \left[45,55\right]$ MHz, N=50, M=50
P4	$f_{c}\in \left[45,55\right]$ MHz, N=50, M=50

**Table 3 sensors-24-05344-t003:** Computational complexity analysis.

Network	CNNQu	CNNHuang	ResNet50	MobileNetV3	ShuffleNetV2	CSANet
FLOPs/G	0.0334	0.0371	4.1317	0.3118	0.4034	0.3386
Params/M	2.5539	21.0068	23.5326	4.2200	2.4930	0.2152
Runtime/ms	4.8750	5.0065	35.6344	18.1344	10.1563	7.6219

**Table 4 sensors-24-05344-t004:** Accuracies of different CSANet architectures.

Network	SNR = −14 dB	SNR = −12 dB	SNR = −10 dB	FLOPs
D-SIF	60.79%	84.53%	94.21%	0.2142 G
D-CCI	76.91%	90.96%	95.40%	0.2546 G
CBAM	57.73%	83.48%	92.91%	0.1823 G
CSA = 1	66.90%	87.11%	97.25%	0.2127 G
CSA = 2	77.14%	92.04%	97.24%	0.3134 G
CSA = 3	83.62%	93.99%	98.23%	0.3386 G
CSA = 4	75.91%	92.38%	96.94%	0.3449 G

## Data Availability

The original contributions presented in the study are included in the article, further inquiries can be directed to the corresponding author.

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
