# Peer review of "Intra-Pulse Modulation Recognition of Radar Signals Based on Efficient Cross-Scale Aware Network"

_sensors, 2024, doi:10.3390/s24165344_

Round 1
Reviewer 1 Report
Comments and Suggestions for Authors
In this paper, authors proposed a cross-scale aware network (CSANet) to improve the accuracy of intra-pulse modulation recognition in low SNR scenarios.
1. I recommend that the authors carefully proof-read the whole paper in order to avoid typos and examples of English misuse.
2. The novelty of the work should be highlighted more explicitly. Is the time-frequency feature fusion method superior to the time-frequency enhancement method[1-2]? Is the classification network superior to existing advanced methods[3-4]? I recommend adding a section that contrasts the proposed method with existing approaches, clearly outlining the novel aspects and the advancements made over prior work.
3 While the paper presents experiments demonstrating the effectiveness of CSANet across twelve radar signal datasets, more details are needed regarding the experimental setup. Specifically, the selection criteria for datasets, the parameter settings for competing methods, and the statistical significance of the results should be included. Additionally, the computational complexity and runtime performance of CSANet compared to other methods should be discussed to substantiate claims of its efficiency and lightweight nature.
Automatic Modulation Classification of Radar Signals Utilizing X-net. Digital Signal Processing.
Comments on the Quality of English LanguageI recommend that the authors carefully proof-read the whole paper in order to avoid typos and examples of English misuse.
Reviewer 2 Report
Comments and Suggestions for Authors
please check the attachment

-
Reviewer 3 Report
Comments and Suggestions for Authors
The paper is devoted to an important problem of the image's recognition with low signal-to-noise ratio. I have some questions and recommendations to the authors.
1) Table 1 containing the radar modulation formulas should be removed or rewritten: this table is poorly readable. I think that it would be better for the authors to describe briefly the most frequently used radar modulation formulas. These models can be summarized in one table but such a table should be representative. All the designations should be clarified.
2) The structure diagram (Fig.1) is not clear. All the abbreviations should be explained. Some more comments to this diagram are needed, in my opinion.
3) Fig.2 also looks rather unreadable. What happens if other values of SNR (less or more than 10 dB) are considered?
4) What are the limits of the integrals in (2), (3) and (5)?
5) The example of implementing the FSST technique shown in Fig.3 should be clarified. I can not see any consentration of the signal in this Figure.
6) The phrase in Line 158 needs a correction: "... SPWD FSST HHS SHOWS" - (show, may be ?)
7) Fig.6 is overloaded with details, in my opinion. Therefore, it is hard to understand how this module operates.
8) The phrase in Line 216 needs a correction: "... needs to obtaineS...".
9) All the abbreviations in Eq. (9)-(10) should be clarified.
10) The quality of Fig. 9 does not allow to recognize how the algorithm proposed by the authors works.
11) The phrase in Line 322 should be corrected: "results... HELPS".
12) Fig. 10 should be supplemented with some comments.
Comments on the Quality of English LanguageEnglish Language needs a moderate revision.
Round 2
Reviewer 1 Report
Comments and Suggestions for Authors
Please add the comparison to the manuscript.
Reviewer 2 Report
Comments and Suggestions for Authors
This article proposes a novel method for the recognition of intra-pulse modulation of radar signals. The author has responded most of the comments well, but there are still some issues that need to be addressed.
1. In section 3, are the parameters of the analyzed radar signals inconsistent in the examples of the three time-frequency analysis methods (Figure 2 - 4)? Using consistent radar parameters can better highlight the differences among the three methods.
2. In Figure 6, there are some parts that need improvement. In subgraph (a), the output size of the last Conv block is not aligned. In subgraph (c), the output of DDConv Group needs to be represented using the same method as the input of CCI and SIF in subgraph (d). In subgraph (c) - (f), although the text provides image explanations, the label positions of some operations and blocks is confusing (such as Hadamard product)
3. In the SIF structure, pooling between channels is adopted, which is different from traditional GAP and GMP. But traditional GAP is used in the structure of CCI, which can cause confusion. It is best to make naming distinctions.
Comments on the Quality of English LanguageMinor editing of English language required
